Polymorphism of amyloid-like fibrils can be defined by the concentration of seeds

Sneideris Tomas
Milto Katažyna
Smirnovas Vytautas vytautas.smirnovas@bti.vu.lt
Department of Biothermodynamics and Drug Design, Vilnius University, Institute of Biotechnology , Vilnius , Lithuania
Uversky Vladimir
Electronic publication date: 2015 Aug 20
Publication date: 2015
Volume: 3
Electronic Location ID: e1207
Received 2015 May 19; Accepted 2015 Aug 1
Copyright: © 2015 Sneideris et al.
Copyright year: 2015
Copyright holder: Sneideris et al.
License: This is an open access article distributed under the terms of the Creative Commons Attribution License, which permits unrestricted use, distribution, reproduction and adaptation in any medium and for any purpose provided that it is properly attributed. For attribution, the original author(s), title, publication source (PeerJ) and either DOI or URL of the article must be cited.
License URL: https://creativecommons.org/licenses/by/4.0/

Keywords: Amyloid, Prion, Protein misfolding, Protein aggregation, Amyloid-like fibrils, Prion strain, Polymorphism, Elongation, Nucleation

Funding: Research Council of Lithuania MIP-030/2012 Marie Curie Career Integration Grant 293476 This research was funded by a grant (No. MIP-030/2012) from the Research Council of Lithuania. VS was supported by a Marie Curie Career Integration Grant 293476. The funders had no role in study design, data collection and analysis, decision to publish, or preparation of the manuscript.

==============================
Prions are infectious proteins where the same protein may express distinct strains. The strains are enciphered by different misfolded conformations. Strain-like phenomena have also been reported in a number of other amyloid-forming proteins. One of the features of amyloid strains is the ability to self-propagate, maintaining a constant set of physical properties despite being propagated under conditions different from those that allowed initial formation of the strain. Here we report a cross-seeding experiment using strains formed under different conditions. Using high concentrations of seeds results in rapid elongation and new fibrils preserve the properties of the seeding fibrils. At low seed concentrations, secondary nucleation plays the major role and new fibrils gain properties predicted by the environment rather than the structure of the seeds. Our findings could explain conformational switching between amyloid strains observed in a wide variety of in vivo and in vitro experiments.

Introduction

Prions are infectious particles which play the main role in a group of fatal neurodegenerative disorders, also known as the transmissible spongiform encephalopaties (TSE’s). Prion diseases propagate by self-replication of a pathogenic prion isoform (PrPSc) using cellular prion protein (PrPC) as a substrate (Prusiner, 1998; Collinge, 2001). Although structures of infectious forms of PrP are still only partially defined, it is known that PrPSc is rich in beta-sheet structure and demonstrates fibrillar morphology (Sim & Caughey, 2009; Colby & Prusiner, 2011). Different conformations of PrPSc are responsible for variations in prion disease phenotypes and are usually referred to as strains (Safar et al., 1998). For a long time, prion protein was the only suspected infective protein in humans; however, recently there is growing evidence that proteins in other amyloid-related diseases may spread via prion-like mechanisms (Lundmark et al., 2002; Soto, Estrada & Castilla, 2006; Frost & Diamond, 2010; Brundin, Melki & Kopito, 2010; Eisele et al., 2010; Angot et al., 2010; Westermark & Westermark, 2010; Masuda-Suzukake et al., 2013; Eisele, 2013; Goedert et al., 2014). Moreover, the most recent data suggest that variants of Alzheimer’s disease are encoded by different strains (Stöhr et al., 2014; Watts et al., 2014; Aguzzi, 2014).

A lot of information on possible mechanisms of amyloid-like fibril formation comes from in vitro studies of the aggregation kinetics (Knowles et al., 2009; Arosio et al., 2014; Meisl et al., 2014). It is thought that four major steps are involved in fibril formation (Meisl et al., 2014). In the case of spontaneous aggregation, everything starts from primary nucleation. It takes time for a group of soluble protein molecules to get together and misfold into an amyloid-like structure, which serves as a nucleus for fibrillation. Once nuclei are formed, they start elongation into fibrils by attaching soluble protein at the ends and refolding it into an amyloid-like structure. Although nucleation and elongation could be sufficient for describing fibrillation, in many cases secondary processes, such as fibril fragmentation and secondary nucleation are extremely important (Knowles et al., 2009; Meisl et al., 2014). Fibril fragmentation increases the number of fibril ends, which leads to faster elongation. The presence of fibrils can induce formation of new nuclei with much shorter lag times compared to primary nucleation; this is referred to as secondary nucleation (Meisl et al., 2014).

How would such a mechanism of fibril formation work in the case of different amyloid strains? Strain-like structural polymorphism was observed in a number of different amyloid-forming proteins (Tanaka et al., 2004; Tanaka et al., 2005; Yamaguchi et al., 2004; Dzwolak et al., 2004; Petkova et al., 2005; Jones & Surewicz, 2005; Heise et al., 2005; Paravastu et al., 2008; Makarava et al., 2009; Colby et al., 2009; Dinkel et al., 2011; Jones et al., 2011; Chatani et al., 2012; Bousset et al., 2013; Ghaemmaghami et al., 2013; Cobb et al., 2014; Tycko, 2014; Surmacz-Chwedoruk, Babenko & Dzwolak, 2014). To form different amyloid strains de novo using the same protein, different environmental conditions, such as temperature (Tanaka et al., 2005), shear forces (Makarava et al., 2009), concentration of denaturants (Cobb et al., 2014) or co-solvents (Dzwolak et al., 2004) are involved. Once nuclei are formed, they are able to carry strain-specific properties even in unfavorable environments (Dzwolak et al., 2004; Petkova et al., 2005; Makarava et al., 2009; Cobb et al., 2014; Surmacz-Chwedoruk, Babenko & Dzwolak, 2014). This indicates that environment defines different strains during primary nucleation, but affects only kinetics, not the structure, of fibrils formed via elongation. In the case of secondary nucleation, formation of new nuclei is induced by existing fibrils, but there is no experimental evidence if the structure of these nuclei is determined by the environment conditions, or by structure of the fibrils. Or in other words, can secondary nucleation be responsible for conformational switching in amyloid-like fibril strains?

Materials and Methods

Recombinant mouse prion protein fragment (rMoPrP(89-230)) used in this study was purified and stored as described previously (Milto, Michailova & Smirnovas, 2014). Protein grade guanidine hydrochloride (GuHCl) was purchased from Carl Roth GmbH, guanidine thiocyanate (GuSCN) and other chemicals were purchased from Fisher Scientific UK.

To prepare different fibril strains, monomeric protein from a stock solution was diluted to a concentration of 0.5 mg/mL in 50 mM phosphate buffer (pH 6) containing 2 M or 4 M GuHCl, and incubated for one week at 37 °C with 220 rpm shaking (in shaker incubator IKA KS 4000i). For seeding experiments rPrP-A4M fibrils were treated for 10 min using Bandelin Sonopuls 3100 ultrasonic homogenizer equipped with MS72 tip (using 20% power, cycles of 30 s/30 s sonication/rest, total energy applied to the sample per cycle—0.36 kJ). The sample was kept on ice during the sonication. Right after the treatment, fibrils were mixed with 0.5 mg/ml of mouse prion solution in 2 M GuHCl in 50 mM phosphate buffer, pH 6, containing 50 µM ThT. Elongation kinetics at 60 °C temperature was monitored by ThT fluorescence assay (excitation at 470 nm, emission at 510 nm) using Qiagen Rotor-Gene Q real-time analyzer (Milto, Michailova & Smirnovas, 2014). ThT fluorescence curves were normalized by dividing each point by the maximum intensity of the curve.

For denaturation assays, amyloid fibrils were resuspended to a concentration of 25 µM in 50 mM phosphate buffer, pH 6, containing 0.5 M GuSCN and homogenized by sonication (same way as in preparation of seeds). These solutions were diluted 1:4 in a buffer containing varying concentrations of GuSCN, and incubated for 60 min at 25 °C in Maxymum Recovery™ microtubes (Axygen Scientific, Inc., Union City, California, USA). 150 µL of samples were mixed with 850 µL of 100 mM phosphate buffer, pH 7, containing ThT (final concentration after dilution was 50 µM), then each mixture was sonicated for 15 s (same conditions as described above). Fluorescence was measured at 480 nm using the excitation wavelength of 440 nm. Denaturation curves were normalized by dividing each point by the average intensity of the points in the plateau region. Fractional loss of signal at increasing denaturant concentrations corresponds to the fraction of rPrP dissociated from amyloid fibrils.

For AFM experiments, 30 µL of the sample were deposited on freshly cleaved mica and left to adsorb for 1 min, the sample was rinsed with several mL of water and dried gently using airflow. AFM images were recorded in the Tapping-in-Air mode at a drive frequency of approximately 300 kHz, using a Dimension Icon (Bruker, Santa Barbara, California, USA) scanning probe microscope system. Aluminium-coated silicon tips (RTESPA-300) from Bruker were used as a probe.

To prepare samples for the FTIR measurements, rMoPrP aggregates were separated from the buffer by centrifugation (30 min, 15,000 g), and resuspended in D2O, sedimentation and resuspension was repeated three times to minimize the amount of GuHCl and H2O. After resuspension samples were homogenized by 1 min sonication (same conditions as described above). The FTIR spectra were recorded using Bruker Alpha spectrometer equipped with deuterium triglycine sulfate (DTGS) detector. For all measurements, CaF2 transmission windows and 0.1 mm Teflon spacers were used. Spectra were recorded at room temperature. For each spectrum, 256 interferograms of 2 cm−1 resolution were co-added. A corresponding buffer spectrum was subtracted from each sample spectrum. All the spectra were normalized to the same area of amide I/I’ band. All data processing was performed using GRAMS software.

Results

Conformational stability of PrPSc as defined by resistance to chemical denaturation has been one of the key parameters used to define differences between strains (Colby et al., 2009). Different strains of recombinant mammalian prion protein amyloid-like fibrils made in 2 and 4 M guanidine hydrochloride (rPrP-A2M and rPrP-A4M, respectively) were thoroughly characterized by Surewicz group (Cobb et al., 2014). We used recombinant N-terminally truncated mouse prion protein (rMoPrP(89-230)) to create rPrP-A2M and rPrP-A4M strains of amyloid-like fibrils. Similar to recent data on recombinant human PrP (Cobb et al., 2014), rMoPrP fibrils formed in 2 and 4 M guanidine hydrochloride (GuHCl) have different conformational stability (Fig. 1). Due to the fact that rPrP-A4M fibrils could not be fully denatured using even 7.5 M GuHCl (Cobb et al., 2014), a denaturation assay using a more strongly chaotropic salt, guanidine thiocynate (GuSCN) was performed. Midpoint of denaturation of rPrP-A2M is at ∼1.8 M GuSCN and rPrP-A4M is at ∼3 M GuSCN, respectively. This difference served as a simple, unbiased marker of different strains in further experiments.

Figure 1 Denaturation profiles of rPrP-A2M and rPrP-A4M fibrils in GuSCN reveal different conformational stabilities.

Standard errors calculated from 6 measurements using Student’s t-distribution at P = 0.05.

In our previous work we have described elongation kinetics at different temperatures and GuHCl concentrations, using rPrP-A2M as a seed (Milto, Michailova & Smirnovas, 2014). It was not possible to get reliable data above 2.5 M GuHCl due to depolymerization of rPrP-A2M. Thus only one way cross-seeding is possible for rPrP-A2M and rPrP-A4M strains. We followed cross-seeding kinetics using different concentrations of seeds. As seen in Fig. 2A, five percent seeds led to fast growth of amyloid-like fibrils from the very beginning, suggesting fast fibril elongation. At 1% seed volume (Fig. 2B) elongation is slower, but after some time the rate of aggregation explodes. At a lower concentration of seeds (Fig. 2C) elongation is very slow and the curve looks sigmoidal, as usually seen in case of spontaneous fibrillation; however in absence of seeds no aggregation was detected within the experimental timeframe. Fitting data suggests that the observed process can be attributed to fibril-induced secondary nucleation (see Supplemental Information). The fibril denaturation assay (Fig. 2D) revealed that stability of fibrils formed in the presence of 5% seeds (midpoint at ∼2.9 M GuSCN) is very similar to rPrP-A4M strain, which was used as a seed. At 1% seed volume (Fig. 2E), stability of fibrils is lower (midpoint at ∼2.2 M GuSCN), and at 0.2% of seeds (Fig. 2F) it is the same (midpoint at ∼1.8 M GuSCN) as the rPrP-A2M strain. This allows hypothesizing that fibrils initiated by secondary nucleation do not follow the seeding template, despite using template fibrils as nucleation sites.

Figure 2 Concentration of seeds determines the mechanism of aggregation and stability of the final strain.

Different amounts of rPrP-A4M fibrils (sonicated for 300 s) were added to the solution of rMoPrP, prepared in 2 M GuHCl, 50 mM phosphate buffer, pH6. The kinetics was followed at 60 °C using Thioflavin T (ThT) fluorescence assay, five data repeats at each seed concentration plotted in (A–C). No change of ThT fluorescence was observed in samples without seeds. Denaturation profiles in GuSCN reveal different conformational stabilities of formed fibrils (D–F). Standard errors calculated from 6 measurements using Student’s t-distribution at P = 0.05.

AFM analysis did not reveal any major differences between rPrP-A4M and rPrP-A2M strains (Figs. 3A and 3B). In both cases fibril diameters range from 4 to 16 nm, however thicker fibrils are more often in samples of rPrP-A2M strains. This difference is more obvious when comparing fibrils formed in presence of 5% and 0.2% seeds (Figs. 3C and 3D). The majority of fibrils formed in presence of high amount of seeds are 4–8 nm in diameter, while these formed at low seed concentration are usually 8–16 nm.

Figure 3 AFM images of rMoPrP amyloid-like aggregates.

(A) and (B) show fibrils of rPrP-A4M and rPrP-A2M strains, (C) and (D) show fibrils formed during cross-seeding in the presence of 5% and 0.2% seeds, respectively.

FTIR spectra of rMoPrP amyloid-like fibrils display major band in the amide I/I’ region, corresponding to beta-sheet structure with subtle difference in band frequencies between rPrP-A4M and rPrP-A2M strains (Fig. 4). The spectrum of rPrP-A4M strain is very similar to the spectrum of fibrils, prepared in the presence of 5% seeds; both show peak maxima at ∼1,620 cm−1. The spectrum of rPrP-A2M strain and the spectrum of fibrils, prepared in the presence of 0.2% seeds show peak maxima at ∼1,624 cm−1. This data serve as additional confirmation that propagation of the strain-specific structure depends on the amount of seeds and possibly on the mechanism of aggregation.

Figure 4 FTIR spectra of rPrP amyloid-like fibrils.

In our previous work we have demonstrated the impact of sonication on the elongation kinetics of PrP fibrils (Milto, Michailova & Smirnovas, 2014). Comparison of microscopy data before (Fig. 3A) and after (Fig. 5) sonication suggests that the main effect of sonication is breaking fibrils into shorter pieces, thus increasing number of fibril ends.

Figure 5 AFM images of rPrP-A4M fibrils after 300 s (A) and 30 s (B) sonication.

As seen in Figs. 6A and 6B, in case of shorter (or in the absence of) sonication, kinetic curves have sigmoidal shapes, similar as in case of lower amount of longer-sonicated seeds. Fibrils formed in the presence of 30 s sonicated seeds (Fig. 6C) are more stable (midpoint at ∼2.8 M GuSCN) compared to the fibrils formed in presence of unsonicated seeds (midpoint at ∼2.3 M GuSCN).

Figure 6 Effect of sonication on the kinetics of aggregation (A–B) and stability of formed fibrils (C–D).

The same amount of seeds (5%) was used in all experiments. Five data repeats plotted in (A) and (B). Standard errors calculated from 6 measurements using Student’s t-distribution at P = 0.05.

Discussion

Taken together the data with different seed concentrations (Fig. 2), and sonication times (Fig. 6), show that stability of fibrils is dependent on the kinetics. Different processes in fibril formation leads to the mixture of rPrP-A4M and rPrP-A2M fibril populations in all samples, and different proportions of two strains determine their denaturation profiles. Increase of fibril ends leads to shorter lag times and faster elongation rates, and to the bigger proportion of more stable fibrils. In fact, we cannot exclude the impact of fibril surface as a catalyzer of secondary nucleation. Larger super-structures can be disrupted by sonication thus releasing more fibril surface. We haven’t found large clumps using microscopy, but the possibility of larger aggregates is suggested by the decline of final fluorescence in samples where mild or no sonication was used. Larger aggregates used as seeds grow further and may settle out of solution leading to the decrese of ThT fluorescence.

It is interesting to compare reproducibility of kinetic curves at different conditions. At highest amount of seeds the data is extremely reproducible (Fig. 2A), which is common for fibril elongation reactions. In case of lowest seed concentration (Fig. 2C) it is also relatively good; however it is poor at moderate seed concentration (Fig. 2B) or in case of unsonicated seeds (Fig. 6B). Nucleation is a stochastic event and the reproducibility goes down with the decrease of the monomer concentration. Due to the elongation of fibrils, average monomer concentration available for nucleation in the presence of 1% seeds is lower than in case of 0.2% seeds, thus it can serve as an explanation of the worse reproducibility.

Amyloid strain switching has been observed in animal studies (Bartz et al., 2000; Asante et al., 2002; Lloyd et al., 2004; Ghaemmaghami et al., 2013), cell culture (Li et al., 2010), and experiments in vitro (Castilla et al., 2008; Makarava et al., 2009; Surmacz-Chwedoruk, Babenko & Dzwolak, 2014). Two possibilities are suggested to explain this phenomenon (Collinge & Clarke, 2007; Cobb & Surewicz, 2009). The first one describes coexistence of multiple structures in the infective material, when only the dominant type would be recognized experimentally; however, upon transmission to different host, the minor population may self-propagate much better and become dominant, reflected in the change of strain properties. Recently this way of amyloid strain switching was demonstrated for insulin fibrils in vitro (Surmacz-Chwedoruk, Babenko & Dzwolak, 2014). The second possibility suggested that sometimes host protein can adopt amyloid conformations distinct from the heterologous template. The Baskakov group demonstrated adaptive conformational switching within individual fibrils as a possible mechanism for such change (Makarava et al., 2009). Our data suggests a possibility of strain switching via secondary nucleation pathways. Moreover, secondary nucleation could explain switching of strains in absence of species barrier, for example in case of recently described Darwinian evolution of prions in cell culture, which showed strain mutations within a single host protein (Li et al., 2010) or in case of protein misfolding cyclic amplification (PMCA) of recombinant PrP (Smirnovas et al., 2009). In summary, we hypothesize that continuous propagation or switching between amyloid strains may be determined by the mechanism of replication in addition to the environment. In cases when a species barrier or environmental barrier stops or slows down fibril elongation, there is the possibility of secondary nucleation events to seed the formation of different strains. The mechanism is dependent on the concentration of fibrils, which opens up a new dimension in cross-species and cross-environment seeding/infection experiments.

We would like to acknowledge that part of the described kinetic profiles differs from the general fibrillation kinetics, normally observed in the field. Thus, in the absence of supporting investigations of different systems, we would like to stress that all our findings may be limited to the described system and any extrapolation to other amyloid proteins and/or other conditions of fibrillation needs an additional experimental evidence.

Supplemental Information

Supplemental Information 1 Curve fitting

Click here for additional data file.

Supplemental Information 2 Fibril stability

Raw data and processing

Click here for additional data file.

Supplemental Information 3 Kinetics of fibril formation

Raw data and processing

Click here for additional data file.

The authors thank Prof. Witold Surewicz for sharing MoPrP(89-230) plasmid, Dr. Marija Jankunec for help with AFM, and Dr. Jonathan G. Cannon for critical reading of the manuscript.

Additional Information and Declarations

Competing Interests

Author Contributions

The authors declare there are no competing interests.

Tomas Sneideris and Katažyna Milto conceived and designed the experiments, performed the experiments, analyzed the data, prepared figures and/or tables, reviewed drafts of the paper.

Vytautas Smirnovas conceived and designed the experiments, analyzed the data, contributed reagents/materials/analysis tools, wrote the paper, prepared figures and/or tables, reviewed drafts of the paper.

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
