# Peer review of "Polymorphism of amyloid-like fibrils can be defined by the concentration of seeds"

_PeerJ, doi:10.7717/peerj.1207_

## Round 0.1 · original submission · Major Revisions

Please carefully address all critical points raised by both reviewers.

·

Basic reporting

The manuscript describes a study aiming to analyse whether seeding effects the formation of different amyloid strains. In particular the study aims to distinguish between primary and secondary nucleation in the definition of strain/fibril internal structure.
The study is interesting, and the goal is very relevant, and indeed if reaching concluding evidence, would be very important to the field. There are however still some open ends, that disserve investigations, before the suggestions that are stated in the manuscript, are justified. The conclusions are thus rather bold, and could benefit from further investigations.

Experimental design

The experimental setup is very simple, but non-the less informative, and the experiments are thoroughly performed.
The experiments consist of (apart from sample preparation) fluorescence spectroscopy to follow the aggregation kinetics under different experimental conditions, coupled with seeding (seeds formed by sonication). The formed fibrils are denatured, to distinguish between different fibril strains.

Validity of the findings

The conclusions are the following:
A): The amount of 4M seeds determine whether the aggregation kinetics are elongation-driven (in the case of large amounts of seeds where the number of ends is high (also because long sonication times are used)) or secondary-nucleation driven (in the case of the lower amounts of seeds).
B): With a large number of free ends (after significant sonication) the process is elongation driven, and the resulting fibril has the same structure (belongs to same strain) as the template. With a smaller number of free ends (after less sonication) the process is secondary-nucleation driven, and the resulting fibril strain follows the experimental conditions.

The overall conclusion is: secondary nucleation does not result in the strain defined by the template, but rather by the experimental conditions. For elongation it is vice versa.

The latter conclusion is taken too far.
Firstly: for A), the data from denaturation are not provided. These should be added. This is necessary to understand whether also here, there is a difference between he denaturation profiles between those fibrils that display lag-phase, and those that do not.
Most importantly, the denaturation curve must also be provided for the intermediate-% of seeding. Here, a two-phase kinetic profile is evident, hence in principle this suggests that both elongation and secondary-nucleation driven kinetics take place, and hence the denaturation curve should be between the two extremities.

The same goes for case B). Please provide the denaturation kinetics for the 30sec sonication also.

However: The statement that long sonication provides more free ends is plausible, but not certain. Sonication has been shown previously also to disrupt potential super-structures (i.e. higher-level morphologies of fibril arrangement). This means that perhaps the sonication would release more fibril surface – not only more fibril ends. This can perhaps be investigated by TEM, which is suggested to be included.

Secondly, it is important to test the hypothesis for more than one particular system, before stating that secondary nucleation means that the template is not followed. If not including a second system in the investigations, then it is important that it is stressed in the text that the conclusions are only valid for this one system investigated. It is not possible to state that a universal rule for amyloid nucleation/strains has been discovered.
This would evidently never be possible, also not when including more systems, but at least the statement would seem more plausible.

Please also consider why there is such a large difference in the steady-state fluorescence behavior. Why is there a constant final fluorescence value for the 5%/300 sec, but a declining final fluorescence value for the 30s/0sec sonciations (and potentially also for the 1% seeding)? Could this be because ThT-binding is excluded during the formation of complex fibril morphologies. Again, this eludes to the suggested investigation of avialble fibril surface versus available surface ends. Again TEM seems to be the most evident suggestion for the investigation.

The supplementary information includes some relevant information: the reproducibility for highly seeded (5%) solutions is high, and the reproducibility for the lowest % is also relatively good, while the reproducibility for the mixed state is very low. Please include this in the main manuscript, and consider why this phenomenon is observed. Are two different populations observable, and are they formed in different proportions for the different/individual kinetic reactions?
The fitting should not be included in the main text, since only the 5% reactions are adequately fitted.

Another basic problem needs some clarification or consideration.
It has been described in many instances how the aggregation kinetics from fibril formation must be described by BOTH primary nucleation (the initial event) and secondary nucleation. The models by e.g. Knowles’ group (cited in the manuscript) include these phenomena.
If the suggestion by the authors of the current manuscript is valid, then the secondary nucleation event that always takes place during the fibrillation reaction, should in principle be able to lead to a second strain. That is, in the beginning of the reaction, the elongation-driven fibril formation must be assumed to be dominant, but then when the secondary-nucleation-driven fibril formation dominates, it should be observable that a second population of fibrils starts forming. This is because the experimental conditions ARE changing during the kinetics, simply because the solution is depleted from starting material. This should thus lead to two-phase kinetics. Why is this so seldom the case? I hence urge the authors to consider the general plausibility of the statements. Is this in accordance with the general fibrillation kinetics that are normally observed in the field?

Additional comments

It is stressed again that the results are valuable and interesting, only would it be beneficial if some further considerations, and preferably some further experiments, are added.

Reviewer 2 ·

Basic reporting

No comments

Experimental design

No comments

Validity of the findings

No comments

Additional comments

The work by Milto and Co-workers presents a study about cross-seeding experiments using different concentration/sonication of seeds. .

The Authors say that "Using high concentrations of seeds results in rapid elongation and new fibrils preserve the properties of the seeding fibrils. At low seed concentrations secondary nucleation plays the major role and new fibrils gain properties predicted by the environment rather than the structure of the seeds"

As far as I understood the Authors obtain the information about the elongation-driven or secondary nucleation driven features from a fitting. Even if I acknowledge the use of fitting based on models to suggest possible mechanisms, I do not think a fitting alone can unequivocally clarify the mechanism behind the reaction. I actually think that what the Authors call elongation-driven can also be a secondary nucleation driven process where the extremely high amount of seeds (5%) is able to suppress the lag time. So, they probably have both a secondary nucleation process and (maybe) an simple elongation pathway. Can the Authors provide an experimental proof that they have only elongation of pre-existing fibrils and exclude any presence of self-catalytic pathways? This is crucial to validate the central conclusion of the manuscript.

Moreover, Authors also mention in the abstract the "structure of the seeds". What do they know about them? What kind of differences in structures can be highlighted depending on the concentration of the seeds? I think that CD, FTIR coupled with imaging (TEM and AFM) would significantly improve the manuscript.

As a consequence, I think that Authors should provide additional data to unambiguously support their conclusions.

---

## Round 0.2 · accepted · Accept

Thank you for addressing the critical points of the reviewers and for the careful revision.